# Style Content Decomposition-based Data Augmentation for Domain Generalizable Medical Image Segmentation

**Zhiqiang Shen**[1,2,4]                                  XXSZQYY@GMAIL.COM
**Peng Cao**[1,2,*]                                     CAOPENG@CSE.NEU.EDU.CN
**Jinzhu Yang**[1,2]                                  YANGJINZHU@CSE.NEU.EDU.CN
[1] *School of Computer Science and Engineering, Northeastern University, Shenyang, China.*
[2] *Key Laboratory of Intelligent Computing in Medical Image, Ministry of Education, Northeastern University, Shenyang, China*

**Osmar R. Zaiane**[3]                                 ZAIANE@CS.UALBERTA.CA
[3] *Alberta Machine Intelligence Institute, University of Alberta, Edmonton, Canada*

**Zhaolin Chen**[4,5]                                 ZHAOLIN.CHEN@MONASH.EDU
[4] *Department of Data Science & AI, Faculty of Information Technology, Monash University, Melbourne, Australia*
[5] *Monash Biomedical Imaging, Monash University, Melbourne, Australia*

**Editors:** Accepted for publication at MIDL 2026

## Abstract

Due to domain shifts across diverse medical imaging modalities, learned segmentation models often suffer significant performance degradation during deployment. We posit that these domain shifts can be categorized into two main components: 1) **"style" shifts**, referring to global disparities in image properties such as illumination, contrast, and color; and 2) **"content" shifts**, involving local discrepancies in anatomical structures. To address the domain shifts in medical image segmentation, we first factorize an image into style codes and content maps, explicitly modeling the "style" and "content" components. Building on this, we introduce a **Sty**le-**Con**tent decomposition-based data **a**ugmentation algorithm (StyCona), which performs augmentation on both the global style and local content of source-domain images, enabling the training of a well-generalized model for domain generalizable medical image segmentation. StyCona is a simple yet effective plug-and-play module that substantially improves model generalization without requiring additional training parameters or modifications to segmentation model architectures. Experiments on cardiac magnetic resonance imaging and fundus photography segmentation tasks, with single and multiple target domains respectively, demonstrate the effectiveness of StyCona and its superiority over state-of-the-art domain generalization methods. The code is available at *https://github.com/Senyh/StyCona*.

**Keywords:** Medical Image Segmentation, Domain Generalization, Style Code, Content Map

---

\* Corresponding Author.

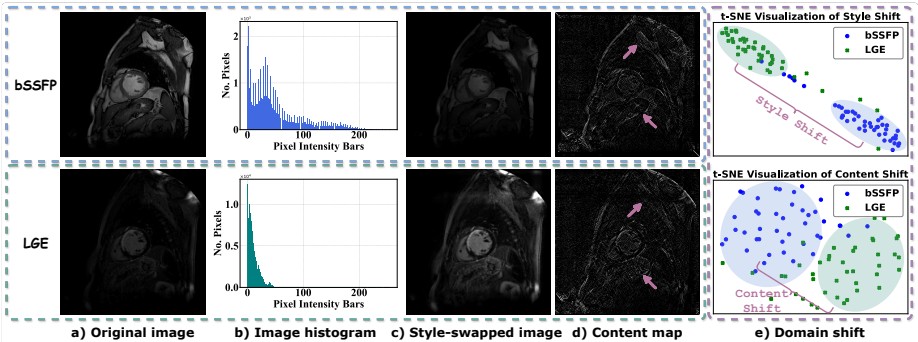

Figure 1: Illustration of style codes and content maps for bSSFP and LGE MRI sequences: a) original images showing disparities in both style (global image appearance) and content (local anatomical structures), b) image histograms reflecting domain shifts in terms of pixel intensities, c) style-swapped images generated by swapping the two original images' style codes (top: bSSFP image with LGE style; bottom: LGE image with bSSFP style), d) content maps of the original images, and e) t-SNE (Van der Maaten and Hinton, 2008) visualization of domain shifts (top: style shift; bottom: content shift).

## 1. Introduction

Medical image segmentation is critical for computer-aided diagnosis. Driven by large and diverse annotated datasets, deep learning-based segmentation models have achieved remarkable progress (Ronneberger et al., 2015; Milletari et al., 2016). However, data distribution mismatches (*a.k.a.*, domain shifts), caused by diverse imaging protocols, equipment vendors, or patient populations, *etc.*, between source (training) and target (testing) domains hinder the generalizability of trained models for clinical deployment (Castro et al., 2020; Zhou et al., 2022; Guan and Liu, 2021).

Domain generalization (DG) aims to mitigate domain shifts by training models on source-domain data and generalizing them to unseen target domains. Existing DG studies generally promote the learning of domain-invariant representations through two major paradigms: representation learning (Chen et al., 2019, 2020; Pei et al., 2021; Peng et al., 2022; Gao et al., 2022, 2023) and data augmentation (*a.k.a.*, domain randomization) (Ouyang et al., 2022; Zhou et al., 2021; Chen et al., 2023, 2025; Gu et al., 2023; Yang and Soatto, 2020; Xu et al., 2021a; Li et al., 2023). The former explicitly explores domain-invariant information via deterministic (Chen et al., 2019, 2020; Pei et al., 2021) or statistical (Gao et al., 2022, 2023) modeling. Although such methods yield intuitive decoupled features on source-domain data, their reliance on domain-specific training often hinders the generalization of disentanglement capabilities across different domains. In contrast, data augmentation-based DG methods alleviate this limitation by expanding the source-domain data distribution and thus implicitly encouraging models to excavate domain-invariant features. As a result, this paradigm has become the mainstream approach in the field. Specifically, these methods typically employ techniques such as Fourier transforma-

tion (Yang and Soatto, 2020; Xu et al., 2021a; Li et al., 2023), random convolution (Xu et al., 2021b; Ouyang et al., 2022; Choi et al., 2023), and feature statistics editing (Zhou et al., 2021; Chen et al., 2023, 2025), *etc* to achieve style augmentation on source-domain data. However, the underlying components involved in medical images remain underexplored. Modeling these components can provide valuable insights into the nature of domain shifts and guide the design of more effective data augmentation strategies.

To bridge this gap, we introduce a style–content decomposition strategy that decomposes an image into *style codes* and *content maps*, revealing that 1) the style code captures global image characteristics within a given anatomical structure and 2) the content maps describe the image's anatomical structure. Fig. 1(c-d) visualizes style-swapped images (obtained by exchanging the style codes between two images) and content maps. It can be observed that the content maps delineate the anatomical structures of the original bSSFP and LGE images, while the style-swapped images exchange appearance between domains but preserve the original anatomical structures. Furthermore, the shift observed in the style codes is more significant than that in the content maps [Fig. 1(e)], as global variations in image characteristics are generally greater than local differences in anatomical structures. Based on these observations, we categorize domain shifts in medical images into two components: 1) **style shifts** (global image property variations) as indicated by deviations in the style codes, and 2) **content shifts** (local anatomical structure discrepancies) as reflected in differences between the content maps. Since quantitatively measuring and reducing these two types of shifts is infeasible without access to target domain data during training, we instead perturb the "style" and "content" components of source domain images to generate augmented images from diverse domains (simulating patients undergoing different imaging systems), enabling the training of a well-generalized medical image segmentation model.

To this end, we propose a **sty**le **con**tent decomposition-based data **a**ugmentation algorithm (**StyCona**) to advance domain generalizable medical image segmentation. Specifically, StyCona performs perturbations on an image's "style" (global image characteristics) and "content" (local anatomical structures) components by 1) blending its style codes and 2) mixing its content maps with those of an auxiliary image, respectively. StyCona generates augmented images with diverse styles and contents while preserving their semantic information, enabling the simulation of images from a wide range of domains for training a well-generalized segmentation model. We evaluate StyCona on cross-domain cardiac magnetic resonance imaging (MRI) segmentation and optic cup (OC)/optic disk (OD) fundus photography segmentation tasks. Experimental results demonstrate that StyCona is a promising solution for domain generalizable medical image segmentation.

Our main contributions can be summarized as follows:

- We propose a style–content decomposition strategy to model the underlying components of medical images and provide a deep insight into domain shifts.

- We propose StyCona, a novel data augmentation algorithm for domain generalizable medical image segmentation. StyCona can be easily integrated into off-the-shelf medical image segmentation backbones and effectively mitigates domain shifts.

- StyCona achieves competitive performance on two domain generalization medical image segmentation benchmarks and provides a compelling alternative to Fourier

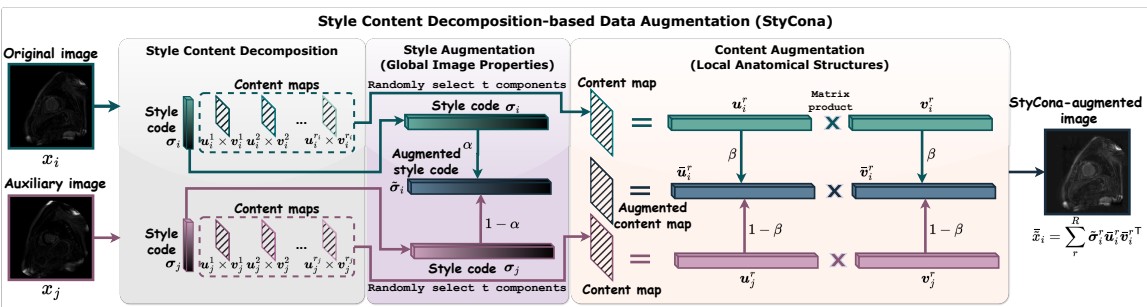

Figure 2: Schematic diagram of the proposed **sty**le **con**tent decomposition-based data **a**ugmentation algorithm (**StyCona**). StyCona includes three steps: 1) style content decomposition, 2) style augmentation, and 3) content augmentation. $x_i$ and $x_j$ represent original images, and $\bar{\tilde{x}}$ denotes a StyCona-augmented sample.

transformation-based, random convolution-based, and feature statistics editing-based domain generalization methods.

## 2. Methodology

For a domain generalizable medical image segmentation task, the training set can be denoted as $\mathcal{D} = \{\mathcal{S}_k | \mathcal{S}_k = \{(x_{i(k)}, y_{i(k)})_{i=1}^{N_k}\}_{k=1}^{K}\}$, where $x_{i(k)}$ represents the $i^{th}$ image in the $k^{th}$ domain and $y_{i(k)}$ is its corresponding ground truth segmentation label. This paper focuses on the most challenging single-source domain generalization setting (*i.e.*, $K = 1$), in which the segmentation model $f(\cdot; \theta)$ is trained on a single source domain and evaluated on one or more unseen target domains. Note that we omit the domain index for brevity.

### 2.1. Style Content Decomposition-based Data Augmentation (StyCona)

Domain shifts in medical images can be characterized two principal components: 1) **"style" shifts**, referring to *global* variations in image properties such as illumination, contrast, and color; and 2) **"content" shifts**, which involve *local* changes in image characteristics and are usually associated with differences in the visibility of specific tissues. Based on this assumption, we propose StyCona to mitigate both style and content shifts through corresponding style and content augmentations. As illustrated in Fig. 2, StyCona mainly includes three steps: 1) style content decomposition, 2) style augmentations, and 3) content augmentation. Specifically, the style-content decomposition is performed using singular value decomposition (SVD) (Klema and Laub, 1980), which factorizes an image into its **singular values (style codes)** and **rank-one matrices (content maps)**. Then, the style and content augmentations are achieved by perturbing the style codes and content maps, respectively.

### 2.1.1. STYLE CONTENT DECOMPOSITION

Low-rank matrices of an image correspond to its principal components (Abdi and Williams, 2010; Candès et al., 2011). Based on this idea, prior work has employed rank-one matrices as the principal parts of an image for restoration tasks (Gao and Zhuang, 2020). We leverage SVD to show that an image's singular values encode its **style codes**, while the associated rank-one matrices represent its **content maps**:

$$x = \sum_{r}^{R} \underbrace{\sigma^r}_{style\,code} \underbrace{\boldsymbol{u}^r \boldsymbol{v}^{r\mathsf{T}}}_{content\,map} \tag{1}$$

where $R$ denotes the rank of the image, corresponding to the number of non-zero singular values. SVD provides the optimal rank-one decomposition in the least-squares sense. It guarantees that the set $\boldsymbol{\Phi} = \{\boldsymbol{u}^1\boldsymbol{v}^{1\mathsf{T}}, \boldsymbol{u}^2\boldsymbol{v}^{2\mathsf{T}}, ..., \boldsymbol{u}^R\boldsymbol{v}^{R\mathsf{T}}\}$ forms a complete basis for the column and row spaces of $x$.

- **Content Map.** Each rank-one matrix in $\boldsymbol{\Phi}$ functions as a basis pattern of the image, and any image structure can be represented as a linear combination of these patterns. Thus, $\boldsymbol{\Phi}$ completely defines the anatomical structures of the image.

- **Style Code.** Each scalar $\sigma^r$ acts as a global multiplier for the corresponding basis pattern $(\boldsymbol{u}^r\boldsymbol{v}^{r\mathsf{T}})$. It modulates the magnitude of pixel intensity in the image, scaling the values of the basis pattern proportionally.

As illustrated in Fig. 1, swapping the style codes of the bSSFP and LGE images results in style transfer between them, while summing all content maps of one of the images reconstructs its complete anatomical structure.

### 2.1.2. STYLE AUGMENTATION

Style augmentation aims to alleviate style shifts (global variations in image properties). As depicted in Fig. 2, we perform style augmentation on an image $x_i$ by blending its **style codes** with those of an auxiliary image $x_j$ in an element-wise manner:

$$\tilde{\sigma}_i^r = \alpha \times \sigma_i^r + (1 - \alpha) \times \sigma_j^r \tag{2}$$

where the weight $\alpha \sim U(0, 1)$ controls the mix strength, $\sigma_i^r$ represents the $r^{th}$ singular value of image $x_i$, and $i/j$ indicates the sample index $(i \neq j)$.

### 2.1.3. CONTENT AUGMENTATION

We devise content augmentation to mitigate content shifts (local disparities in anatomical structure). This is achieved by mixing a set of $t$ randomly selected **content maps** from an image $x_i$ with another set of $t$ randomly selected content maps from an auxiliary image $x_j$ [Fig. 2]. The perturbation to each content map is formulated as:

$$\bar{\boldsymbol{u}}_i^r = \beta \times \boldsymbol{u}_i^r + (1 - \beta) \times \boldsymbol{u}_j^r \quad or \quad \bar{\boldsymbol{v}}_i^r = \beta \times \boldsymbol{v}_i^r + (1 - \beta) \times \boldsymbol{v}_j^r \tag{3}$$

where the weight $\beta \sim U(0, 1)$ and $\boldsymbol{u}_{i/j}^r / \boldsymbol{v}_{i/j}^r$ represents the left/right singular vector of image $x_{i/j}$.

In a nutshell, StyCona performs style and content augmentation by Eq. (1)(2)(3), and generates an augmented image as $\bar{\bar{x}}_i = \sum_{r=1}^R \tilde{\boldsymbol{\sigma}}_i^r \bar{\boldsymbol{u}}_i^r \bar{\boldsymbol{v}}_i^{r\mathsf{T}}$.

### 2.1.4. Loss Supervision

Afterwards, we formulate the segmentation loss $\mathcal{L}$ for training a domain generalizable model $f(\cdot; \theta)$ based on StyCona augmented images:

$$\mathcal{L} = \frac{1}{N} \sum_{i}^{N} \mathcal{L}_{seg}(f(\bar{\bar{x}}_i, \theta), y_i) \tag{4}$$

where $\mathcal{L}_{seg}$ denotes a segmentation criterion.

## 3. Experiments and Results

**Cross-Domain Cardiac Magnetic Resonance Imaging (MRI) Segmentation** The MS-CMR Dataset (*Single Source and Single Target Domain*) dataset (Zhuang, 2018) contains 45 subjects, each with bSSFP and LGE MRI sequences, along with ground truth annotations for the right ventricle (RV), left ventricle (LV), and myocardium (MYO). All images are normalized to the range $[0, 1]$ and resampled to a uniform resolution of $1.0 \times 1.0$ mm. This experimental setting evaluates a segmentation model's ability to generalize across MRI sequences for cardiac structure segmentation, specifically from LGE to bSSFP and vice versa. For each direction, one sequence (*e.g.*, bSSFP) is treated as the source domain and is divided into training and validation sets with a $8 : 2$ ratio, while another sequence (*e.g.*, LGE) serves as the target domain and is used solely for testing.

**Cross-Domain Optic Cup (OC) and Optic Disc (OD) Fundus Image Segmentation** The Fundus Image Benchmark (*Single Source and Multiple Target Domains*) (Chen et al., 2023) is established based on the ORIGA (Zhang et al., 2010), Drishti-GS (Sivaswamy et al., 2014), REFUGE (Orlando et al., 2020), and RIGA (Almazroa et al., 2018) (Bin-Rushed and Magrabia) datasets. This setting evaluates the model's generalization capability for the joint OC and OD segmentation across multiple target domains. The ORIGA dataset is used as the source domain with the training and validation split following TriD (Chen et al., 2023); the other datasets (BinRushed, Drishti-GS, Magrabia, and REFUGE) act as target domains for cross-domain evaluation.

**Implementation Details.** We conducted the experiments using PyTorch (Paszke et al., 2019) on an NVIDIA A40 GPU with 48G GPU memory. All compared methods were optimized using an AdamW optimizer (Kingma and Ba, 2015) with a fixed learning rate of $1e - 4$ during the 100 training epochs for both the cadiac MRI and fundus image segmentation tasks. U-Net (Ronneberger et al., 2015) was employed as the baseline segmentation model. The combination of cross-entropy and Dice loss (Milletari et al., 2016) was used as the segmentation criterion. All images were resized to $256 \times 256$ for both training and testing, and the predicted labels were rescaled to their original resolutions for evaluation. We set the number of perturbed content maps $t = 16$ (please refer to Section 3.2.2 for further analysis).

### 3.1. Comparison with State of the Arts

We compared StyCona with state-of-the-art DG methods: 1) Fourier transformation-based: amplitude swap (AmpSwap) (Yang and Soatto, 2020), amplitude mixup (AmpMix) (Xu

et al., 2021a), and FMAug (Li et al., 2023)); 2) Feature statistics editing-based: MixStyle (Zhou et al., 2021), TriD (Chen et al., 2023), and ConStyX (Chen et al., 2025); 3) Random convolution-based: RandConv (Xu et al., 2021b), CIDA (Ouyang et al., 2022), and PRand-Conv (Choi et al., 2023). We re-implemented all the compared methods in a unified experimental setup for fair comparison. All the methods are built upon the same baseline segmentation model (U-Net). In general, StyCona sets the state-of-the-art in both settings, as shown by the quantitative results [Table 1 and Table 2] and the qualitative examples [Fig. 3], showcasing its superiority in domain generalizable medical image segmentation.

### 3.1.1. RESULTS ON CARDIAC MRI

As reported in Table 1, StyCona achieves consistent performance improvements over the compared methods in the cardiac MRI segmentation task. In general, almost all the compared methods surpass the baseline model, indicating their effectiveness in addressing domain shifts. However, it is worth noting that some methods, *e.g.*, AmpMix (Xu et al., 2021a) and TriD (Chen et al., 2023), yield segmentation results inferior to those of the baseline model in the bSSFP $\rightarrow$ LGE scenario. This may result from the fact that the blended amplitude spectrum in AmpMix and the mixed feature statistics in TriD generate style-augmented images that are insufficient for mitigating content shifts, resulting in the models overfitting spurious correlations between the augmented images and their corresponding segmentation labels. In contrast, StyCona demonstrates a clear advantage in handling both style and content shifts by style content augmentation. These results validate the key idea of StyCona and highlight its effectiveness in domain generalizable medical image segmentation.

### 3.1.2. RESULTS ON FUNDUS IMAGE

We further evaluate StyCona in a single-source, multi-target domain setting. As shown in Table 2, all methods generally achieve comparable average performance across the four target domains. However, the two random convolution-based approaches (*i.e.*, CIDA (Ouyang et al., 2022) and PRandConv (Choi et al., 2023)) yield relatively unsatisfactory results. This may be attributed to that the random convolution operations produce augmented images with distorted content that is inconsistent with the corresponding segmentation labels, thus misleading the segmentation models during training. In contrast, StyCona shows more robust performance across the four target domains, obtaining the highest average DSC and competitive ASD. These results further suggest the effectiveness of StyCona in domain generalizable medical image segmentation with multiple target domains.

### 3.1.3. QUALITATIVE RESULTS.

As shown in Fig. 3, StyCona produces segmentation results with more precise object delineation. The improvement can be attributed to StyCona's style and content augmentations, which enable the augmented samples to cover a wide range of unseen domains, thereby allowing the trained model to generate accurate segmentation maps for unseen data. In comparison, methods such as AmpMix (Xu et al., 2021a), MixStyle (Zhou et al., 2021), and PRandConv (Choi et al., 2023) yield relatively unsatisfactory results. AmpMix and MixStyle rely solely on style augmentation by blending amplitude spectra and adjusting

Table 1: Comparison with state-of-the-art methods on the cross-domain cardiac MRI segmentation task (single target domain). The best and second-best results are highlighted in **bold** and underline, respectively.

| Method | bSSFP → LGE | | LGE → bSSFP | |
|---|---|---|---|---|
| | DSC ↑ | ASD ↓ | DSC ↑ | ASD ↓ |
| U-Net (Ronneberger et al., 2015) | 65.97 | 6.55 | 76.91 | 3.41 |
| AmpSwap (Yang and Soatto, 2020) | 70.93 | 6.48 | 78.74 | 3.26 |
| AmpMix (Xu et al., 2021a) | 63.89 | 6.44 | 80.07 | 2.46 |
| FMAug (Li et al., 2023) | 72.46 | 13.67 | 80.54 | 2.57 |
| MixStyle (Zhou et al., 2021) | 69.26 | 10.91 | 80.81 | 2.69 |
| TriD (Chen et al., 2023) | 65.12 | 10.25 | 79.14 | 4.60 |
| ConStyX (Chen et al., 2025) | 68.63 | 7.89 | 81.30 | 2.85 |
| RandConv (Xu et al., 2021b) | 71.90 | 5.65 | 75.17 | 4.27 |
| CIDA (Ouyang et al., 2022) | 69.72 | 8.63 | 72.85 | 4.61 |
| PRandConv (Choi et al., 2023) | 56.61 | 8.75 | 75.57 | 3.38 |
| StyCona (ours) | **73.39** | **4.33** | **81.59** | **2.24** |

Table 2: Comparison with state-of-the-art methods on the fundus image segmentation tasks (Source: ORIGA; Target: REFUGE, Drishti-GS, BinRushed, and Magrabia). The best and second-best results are highlighted in **bold** and underline, respectively.

| Method | BinRushed | | Drishti-GS | | Magrabia | | REFUGE | | Average | |
|---|---|---|---|---|---|---|---|---|---|---|
| | DSC (%) ↑ | ASD ↓ | DSC (%) ↑ | ASD ↓ | DSC (%) ↑ | ASD ↓ | DSC (%) ↑ | ASD ↓ | DSC (%) ↑ | ASD ↓ |
| U-Net (Ronneberger et al., 2015) | 54.82 | 7.07 | 77.99 | 1.16 | 61.19 | 8.10 | 79.49 | 1.89 | 68.37 | 4.56 |
| AmpSwap (Yang and Soatto, 2020) | 63.56 | 4.36 | 78.16 | 1.13 | 66.13 | 4.45 | 81.84 | 1.45 | 72.41 | 2.85 |
| AmpMix (Xu et al., 2021a) | 65.08 | 4.32 | 77.06 | 1.09 | **68.14** | **4.29** | 79.50 | 1.67 | 72.44 | **2.84** |
| FMAug (Li et al., 2023) | 63.79 | 5.03 | 78.34 | 1.04 | 64.33 | 7.28 | **82.17** | **1.29** | 72.16 | 3.66 |
| MixStyle (Zhou et al., 2021) | 63.95 | 4.72 | 80.36 | 1.06 | 63.97 | 6.04 | 81.93 | 1.45 | 72.55 | 3.32 |
| TriD (Chen et al., 2023) | 66.27 | 4.34 | 77.08 | 1.24 | 65.42 | 4.53 | 79.36 | 1.57 | 72.03 | 2.92 |
| ConStyX (Chen et al., 2025) | 62.91 | 5.99 | **80.69** | **1.03** | 64.47 | 7.16 | 80.15 | 1.85 | 72.05 | 4.01 |
| RandConv (Xu et al., 2021b) | 63.75 | 4.70 | 79.35 | 1.13 | 65.61 | 5.07 | 81.52 | 1.46 | 72.56 | 3.09 |
| CIDA (Ouyang et al., 2022) | 53.16 | 11.34 | 78.55 | 5.18 | 51.27 | 14.61 | 68.09 | 6.62 | 62.76 | 8.44 |
| PRandConv (Choi et al., 2023) | 44.60 | 9.84 | 75.23 | 1.48 | 53.34 | 8.68 | 76.61 | 2.57 | 62.44 | 5.64 |
| StyCona (ours) | **67.80** | **3.60** | 78.43 | 1.08 | 65.71 | 4.37 | 80.68 | 1.49 | **73.16** | 2.63 |

the mean and standard deviation of samples, respectively. Meanwhile, the effectiveness of PRandConv is limited by its progressively applied random convolutions, which generate content-distorted images and fail to adequately alleviate variations in anatomical structures. These qualitative results, consistent with the quantitative performance, further suggest the effectiveness of our approach.

## 3.2. Ablation Study

### 3.2.1. Effectiveness of each component

In Table 3, we present ablation experiments to analyze the contributions of each component of StyCona. These experiments were conducted under the cardiac MRI segmentation task. The results reveal a consistent trend: segmentation performance improves as the proposed style and content augmentation components are gradually integrated into our method. Specifically, the baseline model (*i.e.*, U-Net) performs relatively unsatisfactorily on the target domain, as it tends to learn domain-specific decision rules based on source domain data.

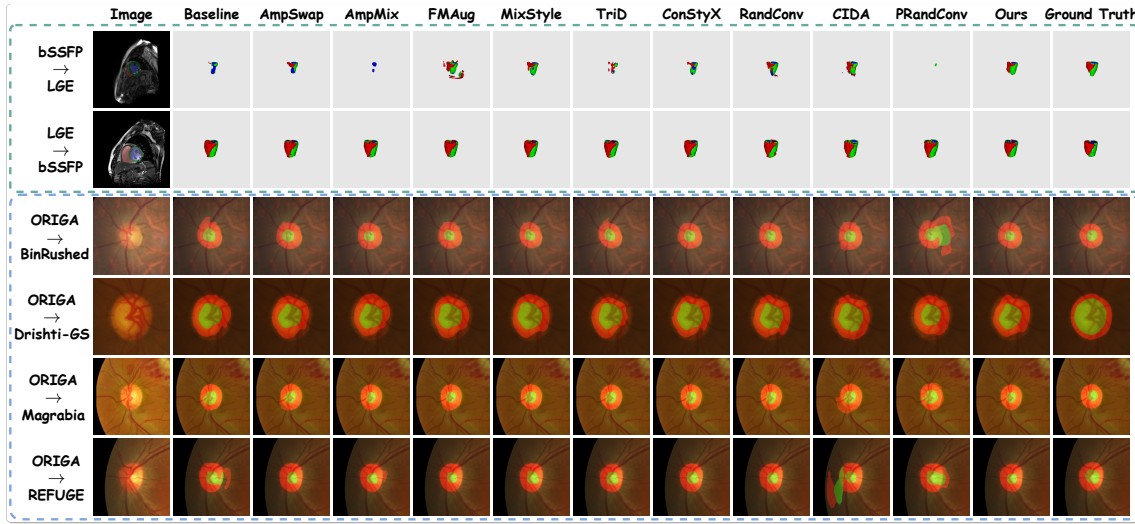

Figure 3: Qualitative results on the cardiac MRI segmentation and the OC/OD fundus image segmentation tasks.

Then, the segmentation performance increases by a large gap when the style augmentation operation is introduced for effectively mitigating style shifts, which are an essential part of domain shifts in these scenarios. In contrast, the improvement is more significant, with an increase of over 7% in DSC, when only the content augmentation component is introduced to address the content shifts, another key aspect of domain shifts. Finally, by incorporating both the style and content augmentation modules to address global variations in image appearance and local differences in anatomical structures, our final method achieves the best performance with DSC of 73.39% and 81.59% on the bSSFP $\rightarrow$ LGE and LGE $\rightarrow$ bSSFP scenarios, respectively. This result demonstrates the effectiveness of each component and validates our assumptions of the style content decomposition and the corresponding style and content augmentation to alleviate the domain shifts.

### 3.2.2. IMPACT OF NO. PERTURBED CONTENT MAPS

According to Eq. (1)(2)(3), perturbing the style codes alters only an image's style without changing its semantic information corresponding to segmentation labels. On the other hand, perturbing $t$ randomly selected content maps results in changes in local anatomical structures (*e.g.*, background tissues or target object-related boundaries), which is the goal of StyCona to enhance the robustness of segmentation models to local anatomical structure variations. Qualitatively, Fig. 4(a) shows the StyCona-augmented samples with varying numbers of perturbed content maps ($t = 8, 16, 32$). Compared with the original images, the augmented images with $t = 8, 16$ exhibit variations in global image appearance and local anatomical structures while remaining consistent with the ground truth segmentation labels. In contrast, excessive perturbation is introduced into some target boundaries when $t = 32$, which may alter their ground truth labels. Quantitatively, $t = 16$ yields the highest

Table 3: Ablation study of StyCona on the cross-sequence setting. The best results are highlighted in **bold**.

| Method | StyCona Style augmentation | Content augmentation | bSSFP → LGE DSC ↑ | ASD ↓ | LGE → bSSFP DSC ↑ | ASD ↓ |
|---|---|---|---|---|---|---|
| Baseline | | | 65.97 | 6.55 | 76.91 | 3.41 |
| Baseline + Style | √ | | 68.98 | 3.20 | 80.45 | 2.40 |
| Baseline + Content | | √ | 72.31 | 5.16 | 81.17 | 2.43 |
| Baseline + Style + Content | √ | √ | **73.39** | **4.33** | **81.59** | **2.24** |

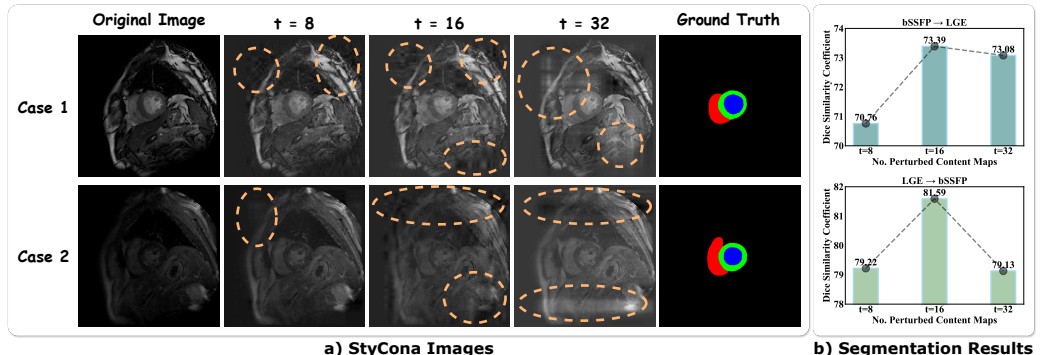

Figure 4: Illustration of a) StyCona-augmented images with different numbers of perturbed content maps ($t = 8$, $t = 16$, and $t = 32$) for two randomly selected original images (The orange dashed circles highlight some content-perturbed regions) and b) Segmentation results for $t = 8$, $t = 16$, and $t = 32$ on cardiac MRI segmentation.

DSC in both the bSSFP → LGE and LGE → bSSFP scenarios. Based on these results, we set $t = 16$ in StyCona to balance perturbation strength with the invariance of segmentation labels, ensuring label consistency after content augmentation.

## 4. Limitations

Despite its effectiveness, StyCona has several limitations that warrant further investigation.

1. **Theoretical understanding of style code and content map.** StyCona adopts SVD to decompose an image into a style code (singular values) and a content map (rank-one matrices). While this design is motivated by empirical observations, a rigorous theoretical proof demonstrating that singular values encode style information and that rank-one matrices capture semantic content is currently unavailable. We consider a deeper theoretical analysis of this decomposition an important direction for future work.

2. **Applicability to diverse image modalities.** StyCona is designed to be modality-agnostic and can be easily applied to different image modalities, including 3D images. However, its effectiveness on various modalities with different characteristics has not

yet been fully explored. We will further evaluate StyCona on more challenging cross-modality medical image segmentation tasks, e.g., CT-MR whole heart segmentation, to explore its limitations.

## 5. Related Work

### 5.1. Domain Generalization

Domain generalization (DG) aims to train a model on a source domain and enable it to generalize to unseen target domains. Numerous DG methods have been proposed for both natural and medical image scenarios (Zhou et al., 2022; Castro et al., 2020). Specifically, these methods enforce the learning of domain-invariant representations through various approaches, including: deterministic or statistical modeling of domain-invariant and domain-specific features (Gu et al., 2023; Gao et al., 2022, 2023); frequency transformation-based mix-up, which combines Fourier amplitude spectra from different images for style augmentation (Yang and Soatto, 2020; Xu et al., 2021a; Li et al., 2023; Zhao et al., 2024); random convolution-based data augmentation, which perturbs pixel intensities via convolution neural networks with random parameters (Xu et al., 2021b; Ouyang et al., 2022; Choi et al., 2023); as well as feature statistics editing-based style transfer, which perturbs image- or feature-level statistics (e.g., mean and standard deviation) to achieve style or content augmentation (Zhou et al., 2021; Chen et al., 2023, 2025). However, the fundamental nature of domain shifts, crucial for guiding the development of more effective DG approaches, remains underexplored. This work first decomposes domain shifts into style and content shifts through modeling the underlying components of medical images and then designs a novel data augmentation algorithm to mitigate both types of domain shifts.

### 5.2. Content-Style Decomposition

Content-style decomposition disentangles features into domain-invariant content and domain-specific style components for addressing the domain shift issue. In the context of semantic segmentation, this technique has been widely employed in the design of both unsupervised domain adaptation (Chen et al., 2019, 2020; Pei et al., 2021; Peng et al., 2022; Yang and Soatto, 2020) and domain generalization (Gu et al., 2023; Gao et al., 2022, 2023; Xu et al., 2021a; Li et al., 2023; Zhao et al., 2024) frameworks. Research in this area generally falls into two categories: 1) learning-based (Chen et al., 2019, 2020; Pei et al., 2021; Peng et al., 2022; Gu et al., 2023; Gao et al., 2022, 2023) and 2) rule-based approaches (Yang and Soatto, 2020; Xu et al., 2021a; Li et al., 2023; Zhao et al., 2024). Learning-based methods rely on additional deep neural network-based encoders and decoders, with specific constraints (e.g., contrastive learning on decoupled components (Gu et al., 2023)), to achieve feature disentanglement either deterministically (Chen et al., 2019, 2020; Pei et al., 2021; Peng et al., 2022; Gu et al., 2023) or statistically (Gao et al., 2022, 2023). Rule-based methods adopt the Fourier transformation to perform direct decomposition. However, they impose the constraint that phase spectra should be fixed to ensure the invariance of the original contents during reconstruction. This constraint limits their flexibility and effectiveness for data augmentation. In contrast, StyCona is a simple yet effective content-style decomposition-based

data augmentation method that factorizes an image into a style code and content maps, enabling style-content augmentation through perturbations on the decoupled components.

## 6. Conclusion

We propose StyCona, a novel data augmentation algorithm for domain generalizable medical image segmentation. Our style content decomposition strategy reveals that an image's singular values function as style codes, governing its global image properties, while its rank-one matrices serve as content maps, determining its anatomical structures. Based on this decomposition, we categorize domain shifts into style and content shifts *w.r.t.* deviations in the style codes and content maps, respectively. Then, StyCona perturbs style codes for style augmentation and blends content maps for content augmentation, addressing both types of domain shifts accordingly. Extensive experiments on cardiac MRI segmentation with a single target domain and fundus image segmentation with multiple target domains demonstrate that StyCona is an effective data augmentation technique for domain generalizable medical image segmentation.

## Acknowledgments

This work was supported by the Science and Technology Joint Project of Liaoning Province (2023JH2/101700367 and ZX20240193), the Fundamental Research Funds for the Central Universities (N25BJD005), the National Natural Science Foundation of China (62076059), and the China Scholarship Council (202406080040).

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
