# OpenReview forum: "Style Content Decomposition-based Data Augmentation for Domain Generalizable Medical Image Segmentation"
_MIDL.io/2026/Conference — MIDL 2026 Poster_

### Official Review · Reviewer_Xxs3 · 2026-01-09

**Confidence:** 4
**Preliminary Rating:** 3

**Summary:**

The paper proposes StyCona, a novel data augmentation algorithm designed to improve domain generalization in medical image segmentation. The authors introduce a "style-content decomposition" strategy based on Singular Value Decomposition (SVD), positing that an image's singular values represent global "style" attributes (illumination, contrast), while the rank-one matrices (singular vectors) represent "content" (anatomical structures). To mitigate domain shifts, StyCona generates augmented training samples by 1) blending the singular values (style) of a source image with those of an auxiliary image, and 2) mixing a subset of rank-one matrices (content) from an auxiliary image into the source image. Experiments on cross-domain cardiac MRI segmentation and multi-target fundus image segmentation demonstrate that StyCona achieves state-of-the-art performance, outperforming recent methods like MixStyle, AmpMix, and RandConv.

**Strengths:**

1. The newly proposed theoretical perspective is interesting. The utilization of SVD to define "style" and "content" is novel. Associating singular values with global intensity modulation is intuitive and well-justified by the visualizations in Figure 1.
2. The proposed StyCona module is parameter-free, requiring no additional learnable weights, and can be easily plugged into standard segmentation architectures like U-Net.
3. The method demonstrates consistent superiority over a comprehensive list of state-of-the-art competitors. The paper also provides a clear ablation study validating the individual contributions of style and content augmentation.

**Weaknesses:**

1. The "content augmentation" strategy involves adding rank-one matrices from a random auxiliary image ($x_j$) to the source image ($x_i$)12. Since medical images are not perfectly registered, the singular vectors of $x_j$ do not spatially align with the anatomy of $x_i$. Blending these components effectively superimposes "ghost" structures or structured noise from a different anatomy onto the training image. Unlike geometric augmentations (which warp labels) or noise injection, this approach risks creating inputs that no longer match the ground truth label $y_i$. The authors rely on a heuristic ($t=16$) to limit this damage13, but this is not a theoretically robust guarantee of semantic preservation.
2. SVD can be a computationally intensive operation ($O(\min(mn^2, m^2n))$), significantly more expensive than the Fast Fourier Transform used in competing methods like AmpMix. The paper does not report the training time overhead introduced by performing SVD and reconstruction for every image in every iteration.
3. The ablation study in Figure 4 shows that increasing the number of perturbed content maps to $t=32$ causes visible distortion and alters ground truth boundaries, leading to performance degradation14. This sensitivity suggests that the method requires careful tuning of $t$ for every new dataset or resolution, limiting its robustness.

**Detailed Comments:**

1. Typo in Equation 1: The term is written as contentmay, which appears to be a typo for "content map" or "content mat"
2. In Table 1, the header LSSFP in the last column seems to be a typo, likely intended to be bSSFP or Average
3. Please clarify if the SVD is applied to the full $256 \times 256$ image or patch-wise. If applied to full images, the computational cost scaling with resolution should be discussed.

**Justification Of The Preliminary Rating:**

The paper presents a clever and effective idea (SVD for DG) with strong experimental results on relevant medical imaging tasks. The novelty of the decomposition strategy is a significant plus. However, the heuristic nature of the "content mixing" step lacks a strong guarantee of label preservation (as admitted by the authors regarding $t=32$), and the absence of computational cost analysis for the SVD operations. Addressing the semantic preservation and efficiency concerns in the rebuttal could raise the score.

**Questions To Address In The Rebuttal:**

1. What is the relative increase in training time (e.g., seconds per epoch) when using StyCona compared to a standard baseline or Fourier-based methods (e.g., AmpMix)?
2. Can you provide a more robust justification (theoretical or empirical) for why mixing singular vectors from unaligned images ($x_j$ to $x_i$) is a valid "content" augmentation rather than just structured noise injection? Did you observe cases where the "ghosting" artifacts negatively impacted convergence?
3. Why are the $t$ content maps selected randomly? Would selecting high-frequency (high index) vs. low-frequency (low index) components have different impacts on the "content shift" simulation?

---

> ### Author Response · Authors · 2026-01-24
> **Response to Reviewer Xxs3**
>
> We thank the reviewer for the very considerable review and insightful comments. The detailed point-by-point responses are given below.
>
>
> Q1. What is the relative increase in training time (e.g., seconds per epoch) when using StyCona compared to a standard baseline or Fourier-based methods (e.g., AmpMix)?
>
> R1. Although SVD is generally associate with a higher theoretical computational complexity than FFT, our StyCona shows comparable training time overhead compared with AmpMix, as image are sparse matrix and using torch.linalg.svd(*, full_matrices=False) can reduce computation cost in practice.
> For example, we have measured the training time overhead on an NVIDIA A40 GPU for the MS-CMR dataset (source domain: bSSPF, target domain: LGE):
> 1) Baseline (no augmentation): 221.3 ms/iteration
> 2) AmpMix (FFT-based): 524.3 ms/iteration
> 3) StyCona (ours): 630.2 ms/iteration
>
> These results indicate that the practical training-time overhead of StyCona remains within an acceptable range for standard domain generalization training pipelines. Our open-source code is available at: https://github.com/Senyh/StyCona.
>
>
> Q2. Can you provide a more robust justification (theoretical or empirical) for why mixing singular vectors from unaligned images (x_j to x_i) is a valid "content" augmentation rather than just structured noise injection? Did you observe cases where the "ghosting" artifacts negatively impacted convergence?
>
> R2. StyCona does not require spatial alignment between original images and auxiliary images. Auxiliary images are randomly sampled from the source domain (i.e., the training set), and no registration or spatial correspondence is assumed or enforced during content mixing. In other words, StyCona does not rely on the spatial alignment of anatomical structures.
>
>
> Q3. Why are the t content maps selected randomly? Would selecting high-frequency (high index) vs. low-frequency (low index) components have different impacts on the "content shift" simulation?
>
> R3. The purpose of content augmentation in StyCona is to simulate local anatomical structure perturbations for enhancing the robustness of segmentation models to local anatomical structure variations.
> Yes, high-frequency (high index) vs. low-frequency (low index) components have different impacts on the "content shift" simulation. Randomly selecting t content maps allows StyCona to introduce diverse content perturbations across multiple spatial scales, rather than biasing the augmentation toward only high-frequency or low-frequency structures. This randomized strategy increases the diversity of anatomical variations in the augmented images and leads to more stable and effective domain generalization performance.
>
>
> Q4. Minor concerns.
>
> R4. We will revise all minor concerns pointed out by the reviewer.

---

### Official Review · Reviewer_y1QE · 2026-01-10

**Confidence:** 4
**Preliminary Rating:** 3
**Final Rating:** 3

**Summary:**

This paper introduces StyCona, a data augmentation method for training domain-generalizable medical image segmentation models. The authors model domain shift as global style changes and local content variations. Using singular value decomposition, each image is decomposed into singular values (interpreted as style codes) and rank-one matrices (interpreted as content maps). StyCona augments training data by blending style codes and content maps across images. The effectiveness of the method is demonstrated on cardiac magnetic resonance imaging and fundus segmentation tasks.

**Strengths:**

- The proposed method is straightforward and intuitive, avoiding unnecessary complexity while remaining mathematically grounded.

- The approach requires no modifications to the underlying architecture, making it a "plug-and-play" solution that is easily adaptable to various segmentation models.

- Despite its simplicity, the method demonstrates competitive performance, often outperforming more computationally heavy or complex state-of-the-art frameworks.

**Weaknesses:**

- The interpretation of singular values as style codes and rank-one matrices as content maps is intuitive but only weakly supported by theoretical analysis. Additional explanation or empirical validation would help clarify why this decomposition consistently captures global appearance and anatomical structure across datasets and modalities.

- Mixing rank-one matrices (content maps) from different images may lead to anatomically implausible structures. While the empirical results demonstrate performance improvements, the paper would benefit from a deeper discussion of potential failure cases of this content mixing strategy, particularly for complex or 3D anatomical structures.

- The experimental evaluation is limited to 2D images with relatively low resolution. This raises concerns regarding the computational cost and scalability of the SVD-based augmentation. Given that most medical imaging data are inherently 3D, the applicability of the proposed method to realistic clinical scenarios remains unclear. A discussion or quantitative comparison of training-time overhead would strengthen the work.

**Detailed Comments:**

- It would be helpful to include a figure illustrating the augmented images. For example, the authors could select two images from the dataset and visualize the resulting augmented images under different values of the α and β parameters. This would help readers better understand the effects of the proposed augmentations.

- There is not any limitations section. It would be great to hear the limitations from the authors and this could help the community.

**Justification Of Final Rating:**

I thank the authors for their clarifications. I strongly believe that a theoretical understanding of the proposed method is important. Additionally, I think that challenging datasets such as CT–MR whole-heart segmentation should be included in this paper rather than left for future work. Since the authors propose a new method, they need to understand and report its basic limitations. Hence, I will keep my score as borderline.

**Justification Of The Preliminary Rating:**

The paper is well written, and the proposed method is simple and effective. However, there are minor concerns regarding its applicability to 3D images and its algorithmic complexity. These aspects should be clarified by the authors.

**Questions To Address In The Rebuttal:**

- What are limitations of this method?

- Most medical imaging modalities (e.g., CT, MRI) are inherently 3D. Is the proposed framework applicable to 3D images? If so, could the authors comment on the computational feasibility and memory overhead of performing SVD on large 3D tensors?

- Could the authors provide further mathematical grounding for the assumption that singular values represent "style codes" and rank-one matrices represent "content maps"? Specifically, how does this interpretation bridge the gap between matrix decomposition and the biological/physical properties of medical images?

- Does the effectiveness of the content-mixing strategy depend on the dataset being spatially aligned (e.g., via registration)? Furthermore, how does the method handle "content mixing" when images depict the same anatomy from significantly different acquisition angles or viewpoints?

---

> ### Author Response · Authors · 2026-01-24
> **Response to Reviewer y1QE**
>
> We thank the reviewer for the very considerable review and insightful comments. The detailed point-by-point responses are given below.
>
>
> Q1. Limitations of StyCona.
>
> R1. As mentioned by the reviewer, despite its effectiveness, StyCona has several limitations that warrant further investigation.
> 1) Theoretical understanding of style code and content map.
> StyCona adopts SVD to decompose an image into a style code (singular values) and a content map (rank-one matrices). While this design is motivated by empirical observations, a rigorous theoretical proof demonstrating that singular values encode style information and that rank-one matrices capture semantic content is currently unavailable. We consider a deeper theoretical analysis of this decomposition an important direction for future work.
> 2) Applicability to diverse image modalities.
> As shown in our open-source code (https://github.com/Senyh/StyCona), StyCona is designed to be modality-agnostic and is easy to be applied to different image modalities, including 3D images. However, its effectiveness on various modalities with different characteristics has not yet been fully explored. We will further evaluate StyCona on more challenging cross-modality medical image segmentation tasks, e.g., CT-MR whole heart segmentation, to explore its limitations.
>
> We will add a new section to discuss the limitations of the proposed method.
>
>
> Q2. Does the effectiveness of the content-mixing strategy depend on the dataset being spatially aligned (e.g., via registration)? Furthermore, how does the method handle "content mixing" when images depict the same anatomy from significantly different acquisition angles or viewpoints?
>
> R2. StyCona does not require spatial alignment between original images and auxiliary images. Auxiliary images are randomly sampled from the source domain (i.e., the training set), and no registration or spatial correspondence is assumed or enforced during content mixing. In other words, StyCona does not rely on the spatial alignment of anatomical structures.
>
>
> Q3. A figure illustrating the augmented images.
>
> R3. We will include a new figure in the final submission to illustrate StyCona-augmented images.

---

### Official Review · Reviewer_Uyx1 · 2026-01-13

**Confidence:** 3
**Preliminary Rating:** 2
**Final Rating:** 3

**Summary:**

The paper proposes to decompose input images into two different components (supposed “style” and “content”) and augment the source domain images with different “style” and “content” from “auxiliary images” representing different domain for the same task.

The method is a classic SVD-based decomposition which does not require learnable parameters and the augmentation happens at the input image level. A linear interpolation of the “style” representated by singular values and “content” represented by singular vectors are performed between source domain images and target domain images (called “auxiliary images”) to generate images that represents wide style.

Such a method is compared against other representative methods showing improved results in cardiac segmentation task using MS-CMR Dataset when generalizing between different MRI sequences (bSSFP and LGE) whereas it is somewhat competitive in the cross-domain fundus imaging datasets.

**Strengths:**

The paper leverages a classic known and widely leveraged method for matrix decomposition, SVD, to decompose and eventually augment target domain features at input level. This "simplicity" which does not require any learnable parameters is a plus.

The paper is well written overall and the results seem promising and atleast competitve to other more complex methods.

**Weaknesses:**

*Additional Robust Evidence*: The paper promises that  “Stycona leads to state-of-the-art performance across wide range of tasks” but the results are somewhat mixed showing somewhat competitive performance in fundus experiment and improved results in the MS-CMR experiment. Additional evidence, confidence intervals and statistical evidence may be required to support the above contribution statement.

*Exploration of why and when the method works*: The results section show that overperturbation of content map hinders performance which is a good start. It would be useful to explore to what extent and in what conditions SVD-based augmentation is able to excel in domain generalizable image-level augmentation?

**Detailed Comments:**

*Figure 2 Clarification*: It is shown in the Figure 2 that two images (original image and auxiliary image) are taken as input but in the datasets section, only source domain images and corresponding groundtruth are used for training and target domain images only for cross-domain evaluation. Could you clearly show which image correspond to source domain image and which correspond to target domain image?

What do the pink arrow indicate in column (d) content map of Figure 1? How were the tsne-plots in column (e) of figure 1 generated?

*SVD-based decomposition in the Literature* May be a literature review of use of SVD for content-style decomposition in other fields.

Is it reasonable to say that the “style is swapped” from one domain to another when the singular values are swapped? Isn’t the singular value also related to their corresponding singular vectors?


typo: left quote marks are backwards. Please use \`\` text '' instead of   ''text '' i.e backquotes`` instead of ''.

**Justification Of Final Rating:**

I am happy to increase the final rating to "borderline" from "weak reject", given some of my and colleagues/reviewers were addressed, especially w.r.t. literature review.

On the other hand, the major concern regarding the proposed method purported as being "leads to state-of-the-art performance" was not strongly backed by evidence (not always the best performing) with statistical significance /confidence interval.  Moreover, the underlying method's property of decomposition of style and content was only weakly supported by theory (as pointed out by fellow reviewer). The proposed method, as is,  is a domain generalization technique that is usually competitive with other methods, but it might be better for the community, where the authors explore insights on when and why certain methods seem to do well in certain settings/datasets.

**Justification Of The Preliminary Rating:**

Given the “simplicity” of the method and the use of classical algorithm for decomposition and competitiveness of the results in some of the presented datasets is a plus, but the proposed SVD-based augmentation method is over-promised as a “leads to state-of-the-art performance across wide range of tasks” which will require validation on additional datasets (more than the current two tasks), confidence interval from multiple runs, statistical significance estimate.
Additionally, exploration of why and when SVD-based augmentation is useful as an enabler of domain-generalizable feature learning could have improved my rating.

**Questions To Address In The Rebuttal:**

*Evidence of SOTA*: “Stycona leads to state-of-the-art performance across wide range of tasks” – this statement might be bit overstretched since , as reported in the fundus images dataset results, it does not always result in sota performance. In three of the target domains (Drishti, Magriba and REFUGE), other methods are able to provide better domain generalization performance.

Also, please add confidence interval to the results, in addition to statistical significance of the proposed method.


*Exploration of why and when the method works*: Could you discuss why the method seem to work well for MS-CMR Dataset (same patient hence more or less same corresponding segmentation?) but other methods seem to do better or competitive in some fundus target domains?
It would be interesting to explore to what extent and in what conditions SVD-based augmentation is able to excel in domain generalizable image-level augmentation? Would it be possible to quantify the content-style separation and content preservation. May be tools for medical imaging dataset comparison tools such as Fréchet Radiomic Distance (FRD) could be useful.

---

> ### Author Response · Authors · 2026-01-24
> **Response to Reviewer Uyx1**
>
> We thank the reviewer for the very considerable review and insightful comments. The detailed point-by-point responses are given below.
>
>
> Q1. Evidence of SOTA.
>
> R1. In the revised manuscript, we will:
> 1) Reduce the contribution claim to state that StyCona achieves competitive performance on two domain generalization medical image segmentation benchmarks.
> 2) Add confidence intervals (mean ± standard deviation) computed over samples.
> 3) Report statistical significance tests (based on paired t-tests) comparing Stycona with the compared methods.
>
>
> Q2. Exploration of why and when the method works.
>
> R2. We were also surprised by the phenomenon that StyCona works better for MS-CMR compared with Fundus images. As shown in our open-source code (https://github.com/Senyh/StyCona), StyCona is designed to be modality-agnostic and can be used for a wide range of image modalities. Moreover, we will further evaluate StyCona on more challenging cross-modality medical image segmentation tasks, e.g., CT-MR whole heart segmentation, to explore its limitations.
> We attribute the performance variance to differences in how the compared methods handle grayscale versus color images. For example, the style transfer-based methods perform relatively poorly on cardiac MRI images due to the lack of “color” variations in the gray-scale MRI, and these methods, in contrast, achieve better performance on fundus images with rich color information.
>
>
> Q3. Figure 2 Clarification.
>
> R3. In Figure 2, both the original image and the auxiliary image are drawn from the source domain, as the target domain is unknown during training. For example, we train a segmentation model on the bSSFP sequence using StyCona for cross-domain data augmentation, and subsequently evaluate the trained model on the LGE sequence.
>
>
> Q4. Figure 1: Pink arrow in column d) and t-SNE in column e).
>
> R4. In Figure 1, the pink arrows highlight the local differences in anatomical structures. The t-SNE visualizations of style shift and content shift are generated by projecting the style codes (singular values) and content maps (rank-one matrices) of images from bSSFP and LGE sequences, respectively.
>
>
> Q5. Literature review on style-content decomposition.
>
> R5. Thank you for the suggestion. We will add a new section “Style Content Decomposition” to review the related literature on style-content decomposition.
>
>
> Q6. Style swap.
>
> R6. As shown in Figure 1, swapping the singular values between two images leads to a corresponding exchange of image styles. This effect is analogous to Fourier transform–based augmentation methods, in which swapping the amplitude spectra results in the transfer of image styles.
>
>
> Q7. Minor concerns.
>
> R7. We will revise all minor concerns pointed out by the reviewer.

---

### Author Rebuttal · Authors · 2026-01-28

**Rebuttal:**

According to the reviewer’s comments, we made two major revisions to the manuscript:
1) adding a discussion of the limitations of StyCona,
2) reviewing the related literature on content–style decomposition.

The corresponding revisions are highlighted in purple in the revised manuscript. Please find the revised manuscript attached.

**Supporting Material:**

/attachment/b0abbd43eeaae27727bef35674a26328325a1c33.pdf

---

### Meta-Review · Area_Chair_3YKB · 2026-02-07

**Recommendation:** Accept (Poster)
**Confidence:** 4

**Metareview:**

This paper proposes StyCona, a simple yet effective SVD-based data augmentation strategy for domain generalization in medical image segmentation. By decomposing images into style and content components and recombining them across domains, the method offers a parameter-free, plug-and-play solution that consistently achieves competitive performance on cross-domain cardiac MRI and fundus segmentation tasks. Although reviewers raised valid concerns regarding the theoretical grounding of the style-content interpretation and robustness of content mixing under imperfect alignment, the method is technically sound and reproducible. Overall, this paper presents a novel and practically useful contribution to domain generalization in medical imaging and is suitable for acceptance.

---

### Decision · Program_Chairs · 2026-02-14

Accept (Poster)